



**Effect of Water Surface Area on the Remotely Sensed Water Quality Parameters of Baysh**
**Dam Lake, Saudi Arabia**
Mohamed Elhag[1*], Ioannis Gitas[2], Anas Othman[1], Jarbou Bahrawi[1]
[1]Department of Hydrology and Water Resources Management, Faculty of Meteorology,
Environment & Arid Land Agriculture, King Abdulaziz University, Jeddah 21589, Saudi Arabia.
[2]Laboratory of Forest Management and Remote Sensing, School of Forestry and Natural
Environment, Aristotle University of Thessaloniki, 54124 Greece.
*Correspondence to: melhag@kau.edu.sa*
**Abstract**
Water quality parameters help to decide the further use of water based on its quality. Changes in
water surface area in the lake shall affect the water quality. Chlorophyll a, Nitrate concentration
and water turbidity were extracted from satellite images to record each variation on these
parameters caused by the water amount in the lake changes. Each water quality measures have
been recorded with its surface area reading to analyses the effects. Water quality parameters were
estimated from Sentinel-2 sensor based on the satellite temporal resolution for the years 2017-
2018. Data were pre-processed then processed to estimate the Maximum Chlorophyll Index (MCI),
Green Normalized Difference Vegetation Index (GNDVI) and Normalized Difference Turbidity
Index (NDTI). The Normalized Difference Water Index (NDWI), was used to calculate and record
the changes in the water surface area in Baysh dam lake. Results showed different correlation
coefficients between the lake surface area and the water quality parameters estimated Remote
Sensing data. The response of the water quality parameters to surface water changes was expressed
in four different surface water categories. MCI is more sensitive to surface water changes rather
than GNDVI and NDTI. Neural network Analysis showed a resemblance between GNDVI and
NDTI expressed in sigmoidal function while MCI showed a different behavior expressed in
exponential behavior. Therefore, monitoring of the surface water area of the lack is essential in
water quality monitoring.
**Keywords:** NDWI, Partition Analysis Water Surface Area, Water Pollution, Water Quality
Monitoring.





## 1. Introduction


Water bodies in lakes and dam's pools exposed to many factors which affect the water quality; the
climate changes disturb the water's temperatures and that lead to increase or decrease the
evaporation rate which play a big role in pollutants' concentrations. The ecosystem of wadi of
Baysh contains considerable amount of vegetations form and large number of trees; in the rainy
season, most of that ecosystem were submerged by water [1, 2].
The organic marital from inside the lake are affecting the water quality. Also, runoff possibly will
transport leaves and wooden pieces to dam's lake as well as the sediment particle which is the
driving force of lake water turbidity [3, 4]. The amount of these organic marital in the lake is
fundamental part of the living organisms in the dam lake including bacteria and algal live cycles
[5].
For seeking the knowledge, losing a 1000 $m^3$ of a fresh water is not a disaster. Nevertheless,
keeping around 140 million $m^3$ of rain water for microorganisms and algae to grow could be a
catastrophe. Through history, small polluted pools were responsible of hundreds of deaths. The
rain water in end of any watershed contains many elements and organic materials [6, 7].
Microorganisms consume the organic material and deoxidized the water. After that, the green algae
start to grow and creating visible layer over the surface, some kinds of these algae generate toxic
gases and pollute all the water body [8].
If the lake at Baysh Dam start to develop such harmful algae colonies on its surface in the presence
of sunlight and shortage of rainfalls, the developing of harmful algae could be uncontrolled and
pollutes the soil and ground water [9].





In some cases, the change in the water quality measures could be minor and unnoticeable, but with
continuity and time the water body will get contaminated and then will affect the ecosystem around
it. Water quality monitoring and pollution prevention are better than having over a 100 million m$^3$
of contaminated water in one location will affects the region. Also, it will need for huge budgets
for the future treatments [10, 11].
Baysh Dam designed to hold 190 million m$^3$ within a surface area of 8 km$^{2;}$ area at full capacity.
The actual surface area never been recorded at full capacity for safety purpose [12]. The maximum
safe operation capacity at Baysh dam is 120 million m$^3$ with surface area of 4.4 km$^2$; at the safe
operational level the surface area rapidly changes with any inflow or outflow from the dam; rapids
change happen due to the shape of the wadi at operational elevation.
The quest for remote sensing applications to monitor water quality parameters  is required to
minimize the human efforts to the lowest level [13].  Sentinel-2 sensor developed by the European
Space Agency (ESA) provides data with high spatial resolution and equipped to practice models
to detect water quality parameters. Most recent study on Sentinel-2 show that the most accurate
algorithm to acquire the highest reflectance for Normalized Difference Water Index (NDWI)
coming from bond 5 and bond 3 [14].
Sentinel-2 bands were used to record the surface area of the lake and to develop a model to detect
chlorophyll and nitrogen concentrations with low root mean squared error [4, 15]. Furthermore,
the selected satellite occupied with multispectral imager MSI which studied and proved in more
than one study to be more accurate than the moderate resolution imaging spectroradiometer. MSI
been used to detect suspended particulate matter in water body and its results were accepted with
wave length  range of  560nm to 780nm.[16].





The main objective in the current study is to monitor the effect of the lake surface area on the water
quality. Maximum Chlorophyll Index (MCI), Green Normalized Difference Vegetation Index
(GNDVI) and Normalized Difference Turbidity Index (NDTI) will be Estimated to represent the
water quality parameters in the dam lake and Normalized Difference Water Index (NDWI) will be
used to delineate the lake surface area. Partition analysis and Artificial Neural Network Analysis
will be used to envisage the water surface area effect on the estimated water quality parameters.
**2. Materials and Methods**
**2.1. Study Area**
Baysh dam is located at the western part of Asir mountains, approximately 100 km north of Jizan
city, Saudi Arabia (Figure 1). The dam is in an arid region with distinguished difference in
temperatures which has a huge effect on Algae growing and oxygen dissolving eutrophication
processes. The dam is constructed for flood control, irrigation of farmland and groundwater
recharge. Also, there is a water treatment plant located about 5 kilo meters from Baysh Dams'
gates. The plant operates in two phases, first phase is the conventional water treatment and the
second phase is the Reverse Osmosis (RO) water treatment plant. The water treatment plant
produces 70000 m$^3$/ daily of irrigating water and been managed and used by the ministry of
Environment water and agriculture. The catchment area of the dam is more than 4000 km$^2$. [3, 6].
The turbidity of the dam lake is acceptable as much as the water volume behind the dam is over
80 million m$^3$ [4]. On the other hand , the water treatment plant requirs low turbidity to operate in
norml mode so the dam's authoroty in Jazan opens the dam gates to lower the water level for safety
sake and not ot decrease it less than 80 milions m$^3$ in order to get low turbidity water for the
treatment plant.






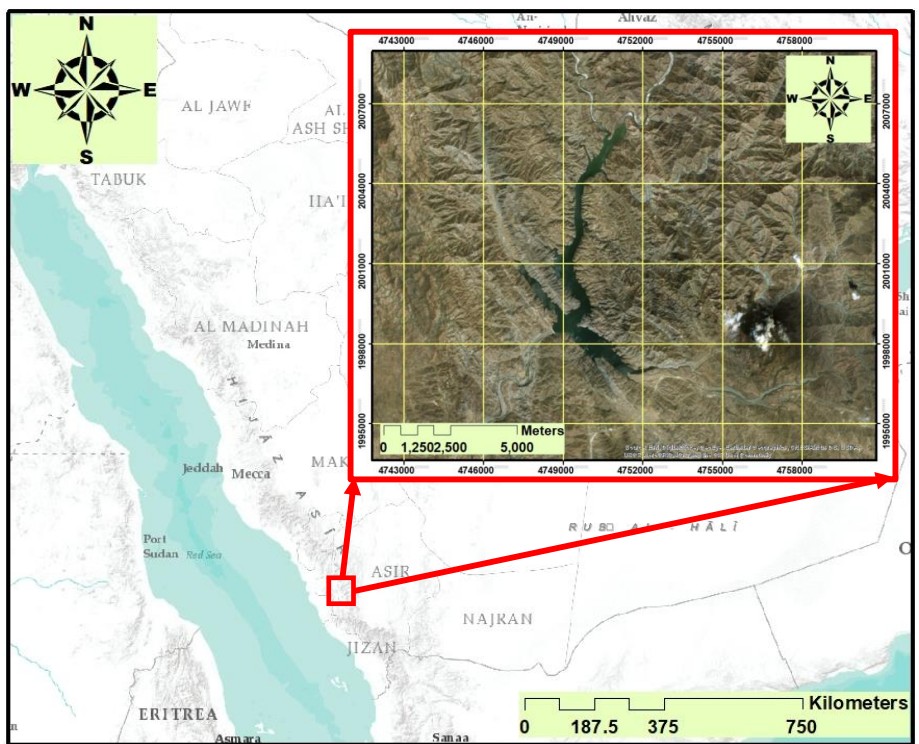

**Figure 1. Location of the Study area [4].**
**2.2. Remote Sensing Data Collection**
Data collection started on January 2017 and last until December 2018 on a temporal resolution of
the satellite instrument which resulted in 52 scenes in total. The sensor is made of 12 spectral
bands, 3-Visible bands (VI) with 10 m resolution, 5-Vegetation Red Edge (VRE) and InfraRed
(IR) bands of 20 m resolution of and 2-Short-Wave InfraRed (SWIR) bands 60 m resolution in
addition to 3 bands related to coastal aerosols and water vapor of 60 m resolution. ESA two levels
of treated images which are 1B and 1C [17]. Level 1C been used in this paper because 1C images
contains radiometric and geometric corrections. The geodetic system for level 1C images is
WGS84 [18].


### 2.3. Realization of Water Quality Parameters

Three different remotely sensed indices were obtained to represent three different water quality parameters, Maximum Chlorophyll Index (MCI), Green Normalized Difference Vegetation Index (GNDVI) and Normalized Difference Turbidity Index (NDTI). The water quality parameters of MCI, GNDVI and NDTI were realized according to Matthews et al. [19], Gitelson and Merzlyak [20] and Lacaux et al. [21] respectively. Detailed exercises of the water quality parameter realizations were discussed in Elhag et al. [4]. While, the Normalized Difference Water Index (NDWI) was found by Gao [22] Then improved by Ganaie et al. [23] to measure the liquid water molecules at the Top Of Canopy (TOC) level. NDWI is calculated by the following equation:

$$NDWI = \frac{NIR - SWIR}{NIR + SWIR}$$
Eq. 1

Where

*NIR* is Sentinel-2 Near InfraRed Band

*SWIS* is Sentinel-2 Short-Wave InfraRed Band

### 2.4. Regression analysis

The regression analysis is the practice of creating a curve, or mathematical function that has the best fit to a series of data points, possibly subject to constraints. There are several fitting functions and there is no general best fit. Best fit is a data dimension and mathematical function dependent [24-27].

In order to describe the effect of the surface area on the water quality parameters at the dam's lake, relations between different surface water area and water quality measures must be examined. The scatter plot been conducted on both variables to visualize the connection between water surface



area and the quality parameters. The readings of the water quality measures are independent
variables, also, the calculated area values are independent. In this case, the Principal Component
Analysis, Neural Network Analysis and Partition Analysis  are the verified methods of exploring
the relation between two independent variables [28, 29].
**2.4.1. Principal Component Analysis**
Principle Component Analysis (PCA) is performed to transform a set of likely correlated with
unlikely correlated variables. Principal components number is less/equal to the variables original
number. Following Monahan [30], PCA fundamental equations are:
First vector $w_{(1)}$ should be answered as follows:
$$w_{(1)} = \arg max_{\|w\|=1} \left\{ \sum_i (t_1)^2_{(i)} \right\} = \arg max_{\|w\|=1} \left\{ \sum_i (x_i . w)^2 \right\} \qquad \text{Eq. 2}$$
The matrix form of the above equation gives the following:
$$w_{(1)} = \arg max_{\|w\|=1} \left\{ \|Xw\|^2 \right\} = \arg max_{\|w\|=1} \left\{ w^T X^T Xw \right\} \qquad \text{Eq. 3}$$
$w_{(1)}$ should be answered as follows:
$$w_{(1)} = \arg max \left\{ \frac{w^T X^T Xw}{w^T w} \right\} \qquad \text{Eq. 4}$$
Originated $w_{(1)}$ suggests that first component of a data vector $x_{(i)}$ can then be expressed as a score
of $t1_{(i)} = x_{(i)} \cdot w_{(1)}$ in the transformed coordinates, or as
the corresponding vector in the original variables, $(x_{(i)} \cdot w_{(1)})\, w_{(1)}$.
**2.4.2. Neural Network Analysis**
The neural network regression model is written as:





$Y = \alpha + \sum_h w_h \phi_h(\alpha_h + \sum_{i=1}^{p} w_{ih} X_i))$         Eq. 5

Where
$Y = E(Y|\boldsymbol{X})$ .
This neural network model has 1 hidden layer, but it is possible to have additional hidden layers.
The $\phi(z)$ function used is hyperbolic tangent activation function. It's used for logistic activation
for the hidden layers.
$\phi(z) = \tanh(z) = \frac{1-e^{-2z}}{1+e^{-2z}}$         Eq. 6

It is significant that the final outputs to be linear not to constrain the predictions to be between 0
and 1. The equation for the skip-layer neural network for regression is shown below:
$Y = \alpha + \sum_{i=1}^{p} \beta_i X_i + \sum_h w_h \phi_h(\alpha_h + \sum_{i=1}^{p} w_{ih} X_i))$         Eq. 7

Cross-validation is therefore critical to make sure that the predictive performance of the neural
network model is adequate. Recall the skip-layer neural network regression model looks like this:
$Y = \alpha + \sum_{i=1}^{p} \beta_i X_i + \sum_h w_h \phi_h(\alpha_h + \sum_{i=1}^{p} w_{ih} X_i))$         Eq. 8

**2.4.3. Partition Analysis**
The partition methods used to contribute all the conditions to main function of this paper. Each
quality parameter in the lake has its own characters and conditions, consequently, the changes in
the surface area affects each parameter in special way which been explained throughout the
partition analysis [31].
Euler invented a generating function which gives rise to a recurrence equation in P(n) Berndt 1994,



$P(n) = \frac{1}{n}\sum_{k=0}^{n-1}\sigma_1(n-k)\,P(k)$                                                      Eq. 9
Where
$\sigma_1(n)$ is the divisor function as well as the identity.
A recurrence relation involving the partition function Q is given by Hirschhorn (1999):
$P(n) = \sum_{k=0}^{[n/2]}Q(n-2k)\,P(k)$                                                      Eq. 10
**3. Results and Discussion**
Changes in the lake's surface area has a clear effect on the dam's water quality. As surface area
and remotely sensed water quality values been collected form satellite images, the relation between
these two is water surface area dependent. Whenever the surface area of the dam's lake changes,
the water quality of the dam lake got affected. Even though, the effect on the MCI values is weak,
but has same inverse relation with surface area [32].
**3.1. Regression analysis**
Regression results showed that mean pixel values were the best to present coherent association
between the water quality parameters and the remotely estimated surface area. Changings in the
surface area effect each water quality parameter in slightly different way. MCI, GNDVI and NDTI
where the main quality parameters in this study. Figure 2 shows a robust correlation of MCI mean
pixel values ($R^2 = 0.94$) with the dam lake surface area in km, also, it clarifies the positive
connections of the MCI mean values. Same processes were conducted on GNDVI and NDTI
values to find and represent the correlation between the variables. $R^2$ for the GNDVI and NDTI
mean values are counted for 0.95 each.





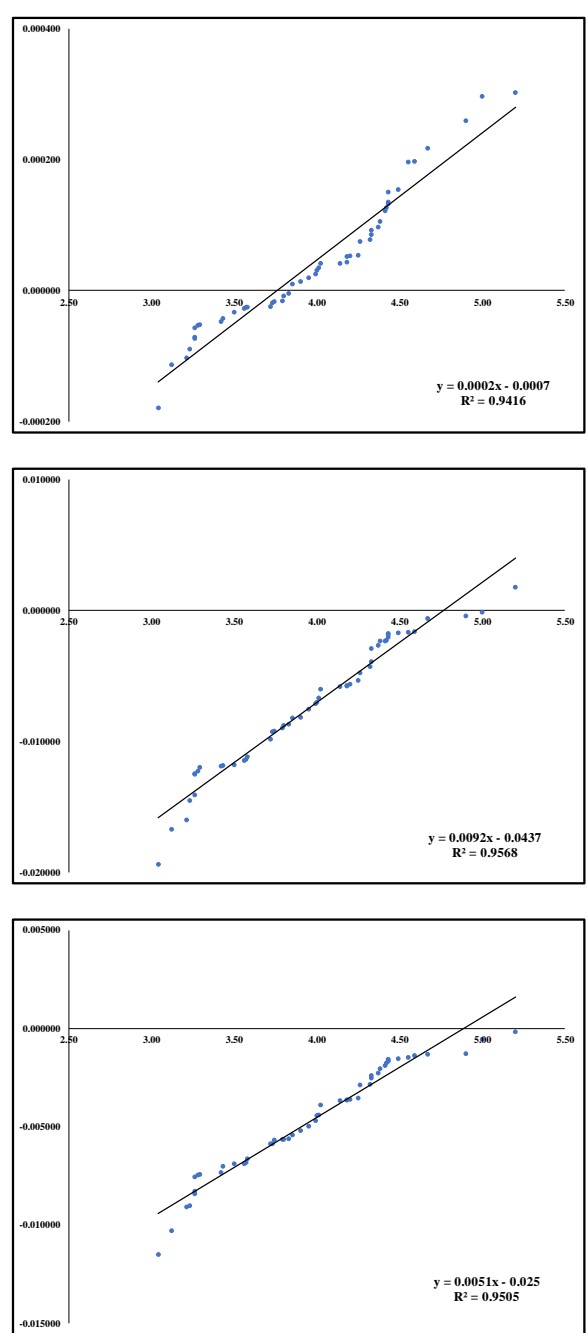

**Figure 2. MCI, GNDVI and NDTI mean pixel value(Y-axis) correlation with the dam lake**

**surface area (X-axis.**







### 3.1.1. Principal Component Analysis


Root Mean Square Error (RMSE) was conducted to confirms the association between the mean
value of the in-situ water quality measurements and the conducted values from remote sensing
data according to summary of fit analysis. the effect of the area change has a clear on NDTI with
very minor on the other components, MCI and GNDVI. But with a separate analysis for each
quality measure, more than 95% of the quality values are responding positively with the decreased
surface areas [33]. The direction and magnitude of the mixed connection between the quality
measures and the change of the surface area are described in Figure 3. The separated analysis of
the quality data could be misguided because of the outlier's numbers. Also, each quality parameter
has its own correlation line, which is different than the other parameter [34, 35].
It's obvious that MCI has its own in response to the dam lake surface area changes rather than
GNDVI and NDTI. This finding is also supported by the Neural Network Analysis showed in
Table 1, where there the prediction profile of the MCI expresses an exponential trend while
GNDVI and DNTI expresses a sigmoidal trend.

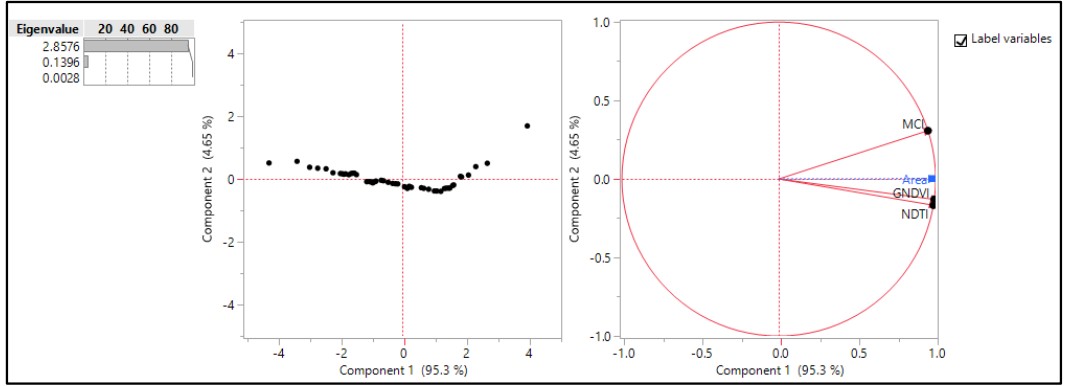


**Figure 3. Principle Component Analysis of the remotely sensed water quality parameters.**





**3.1.2 Neural Network Analysis**
The total number of contributed values which injected in neural network is 51 values using 1
hidden layer and two nodes as shown in Figure 4. The hidden layer on this neural network is
sensitive to the change in surface area. As results for the quality parameters, it is promising results
with very low percentage error.

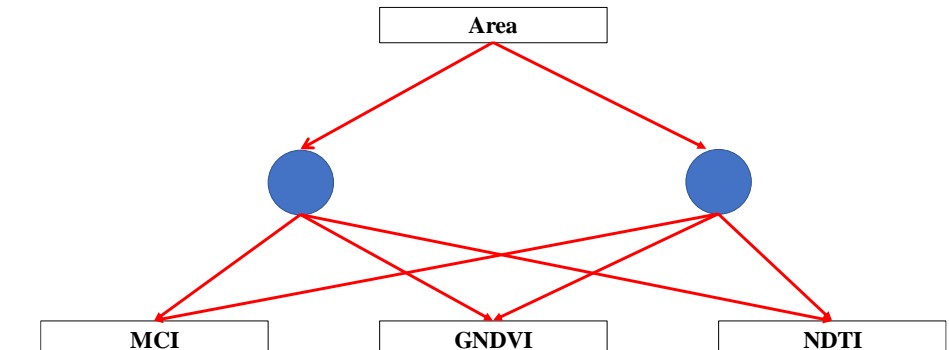


**Figure 4. Neural Network, the interaction of area changes on all parameters as one effect,**
**and the complicate connection between the variables.**
The MCI values has percentage error less than 0.0012%, and the regression line of the points [36]
has $R^2$ value of 0.977. The predicted values of MCI with the measured data generate an exponential
data line which clarify the connection between the water surface area and MCI concentration at
Baysh dam [4, 14].
The sigmoidal function is showed in Table. 1 for the GNDVI and NDTI values, the regressing
lines interact inversely with surface area changes but not in an exponential manner as the
concentration of chlorophyll does. For nitrogen concentration the number of points which used in
this specific Neural network is 34 readings with $R^2$ value of 0.953. The total number of the
Nitrogen reading is 51, but 34 were used to keep 17 values for validation of the results from the





223 neural network. For the water turbidity, $R^2$ for the measured value is 0.95 and for the predicted

224 values is 0.98; for the same parameter RMSE is 0.00026. Validation process for all parameters are

225 presented on Table. 1.

226 **Table 1. Neural Network Analysis for the remotely sensed water quality parameters**

| Model NTanH(2) | | | | Prediction Profiler |
|---|---|---|---|---|
| **MCI** | | | | |
| **Training** | | **Validation** | | |
| **Measure** | **Value** | **Measure** | **Value** | |
| RSquare | 0.9672577 | RSquare | 0.9773973 | |
| RMSE | 2.3554e-5 | RMSE | 1.1815e-5 | |
| Mean Abs Dev | 1.6821e-5 | Mean Abs Dev | 1.0164e-5 | |
| -LogLikelihood | -314.0673 | -LogLikelihood | -168.7629 | |
| SSE | 1.8863e-8 | SSE | 2.373e-9 | |
| Sum Freq | 34 | Sum Freq | 17 | |
| | | | | |
| **GNDVI** | | | | |
| **Training** | | **Validation** | | |
| **Measure** | **Value** | **Measure** | **Value** | |
| RSquare | 0.9533811 | RSquare | 0.9692365 | |
| RMSE | 0.0011226 | RMSE | 0.0006122 | |
| Mean Abs Dev | 0.0007773 | Mean Abs Dev | 0.000501 | |
| -LogLikelihood | -182.6876 | -LogLikelihood | -101.6521 | |
| SSE | 4.2848e-5 | SSE | 6.3711e-6 | |
| Sum Freq | 34 | Sum Freq | 17 | |
| | | | | |
| **NDTI** | | | | |
| **Training** | | **Validation** | | |
| **Measure** | **Value** | **Measure** | **Value** | |
| RSquare | 0.9507846 | RSquare | 0.9824817 | |
| RMSE | 0.0006395 | RMSE | 0.0002676 | |
| Mean Abs Dev | 0.0004335 | Mean Abs Dev | 0.0002 | |
| -LogLikelihood | -201.8175 | -LogLikelihood | -115.7203 | |
| SSE | 0.0000139 | SSE | 1.2174e-6 | |
| Sum Freq | 34 | Sum Freq | 17 | |







### 3.1.3. Partition Analysis


The surface area values were divided into four area levels in order to emphasis on minor changes
in the water quality parameters (Figure 5). The effect on the chlorophyll concentration were minor
because of the interaction with other factors. But the effect is trackable and notable. There are four
splits in the partition analysis based on the LogWorth statistics. The decision tree showed
unevenness in surface area splits affecting Maximum Chlorophyll Index indicating the later
sensitivity to surface area [32, 37]. Surface area of 3.28 km$^2$ has the maximum LogWorth value
(4.99) pointing out the optimal split [37].
Same procedures were conducted on the Green Normalized Difference Vegetation Index and
Normalized Difference Turbidity Index illustrated in Figure 6 and 7 respectively. Although,
GNDVI and NDTI showed decision tree evenness with four splits, but the optimum LogWorth
values counted for 23.7 and 15.63 correspondingly at the same surface area split (3.28 km$^2$). Such
finding supports the vulnerability of nitrogen concertation towards lake surface area changes [38].
Therefore, monitoring dam lake surface area based on the LogWorth statistics is very crucial. In
the current study, lake surface area of 3.28 km$^2$ demonstrated to be critical to the estimated water
quality parameters.

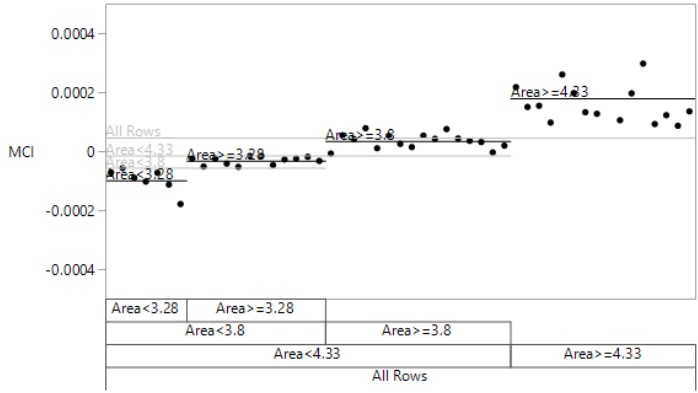






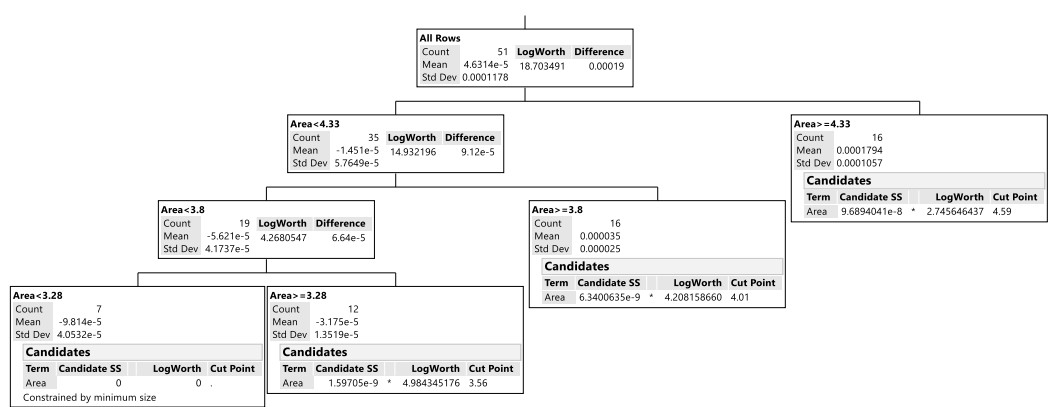


**Figure 5. Decision tree for MCI values with different surface area splits.**


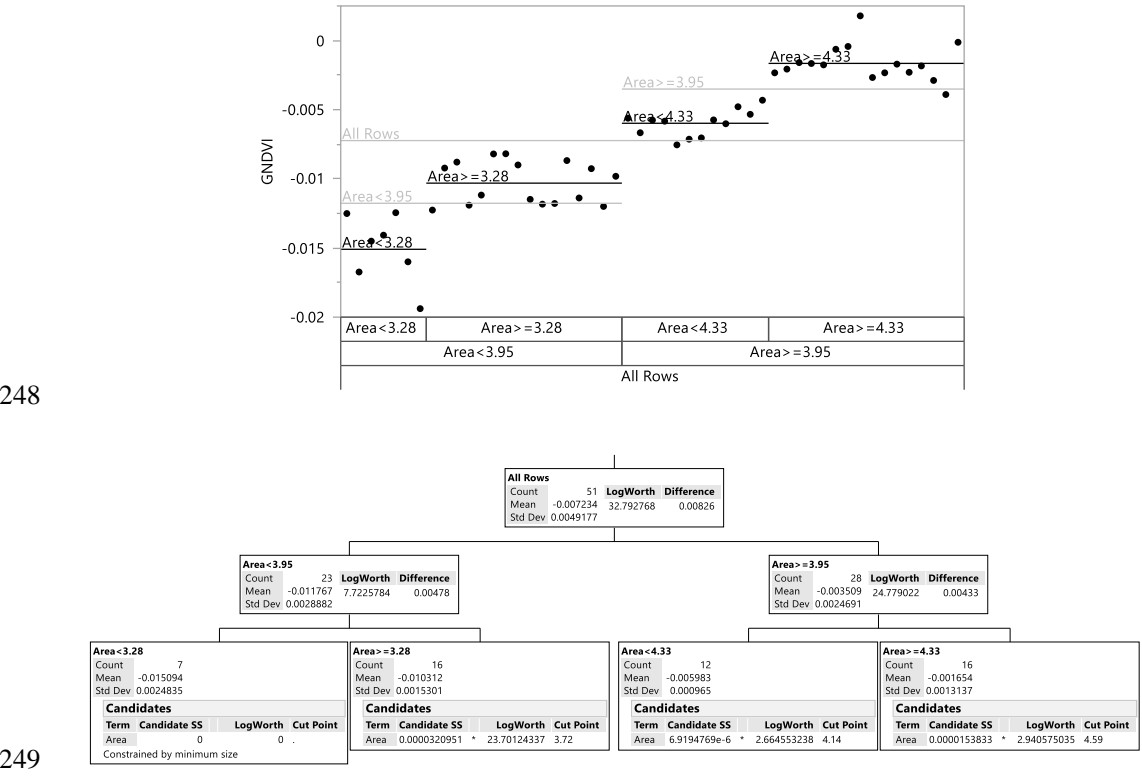




**Figure 6. Decision tree for GNDVI values with different surface area splits.**





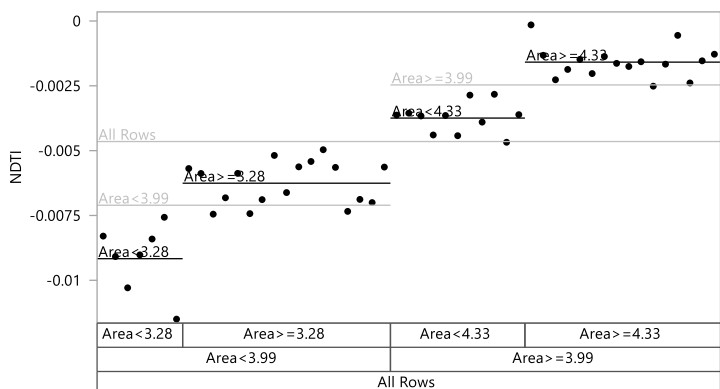



**Figure 7. Decision tree for NDTI values with different surface area splits.**
**4. Conclusions**
Changes in the lake's surface area has a clear effect on the turbidity of the Dam's water. As surface
area and NDTI values been collected form satellite images, the relation between these two is
proportionally related. Whenever the surface area of the dam's lake increases, the turbidity of the
water decreases. Even though, the effect on the MCI values is weak, but has same consistency
relation with surface area. Surface area of the lake surface is supplementary expression of the water
amount in Baysh dam. With the analysis of water quality parameters in the last two years, the
relation between the amount of water which expressed in this study as the water surface area and
the chlorophyll concentration, nitrogen concentration and the sedimentation process is a





corresponding relation. Nevertheless, chlorophyll concertation expressed a sensitive behavior to
changes in the lake surface while nitrogen concentration and turbidity expressed more steady
behavior.
**Acknowledgment**
This project was funded by the Deanship of Scientific Research (DSR) at King Abdulaziz
University, Jeddah, under grant no. **KEP-MSc-01-155-38**. The authors, therefore, acknowledge
with thanks DSR for technical and financial support.
**Author Contributions**
Conceptualization, M.E. and I.Y.; Methodology, M.E. and A.O.; Validation, J.B., M.E.; Formal
Analysis, M.E. and A.O.; Writing – Original Draft Preparation, M.E.; Writing – Review & Editing,
M.E. and A. O.
**Conflicts of Interest**
The authors declare no conflict of interest.

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
