# Peer review of "Effect of Water Surface Area on the Remotely Sensed Water Quality Parameters of Baysh 2 Dam Lake, Saudi Arabia Mohamed Elhag1\*, Ioannis Gitas2, Anas Othman1, Jarbou Bahrawi1 3 4 1Department of Hydrology and Water Resources Management, Fa"

_Hydrology and Earth System Sciences, 2019_

## Referee Comment (RC1) · Silvena Boteva (Referee) · 2 Aug 2019

Comments and Suggestions for Authors

The manuscript entitled "Effect of Water Surface Area on the Remotely Sensed Water Quality Parameters of Baysh Dam Lake, Saudi Arabia" by Elhag et al. presents a study based on the hypothesis that changes in surface area can affect water quality in Baysh Dam Lake. To evaluate this hypothesis, authors have used remotely estimated water surface area, water quality indices, and several statistical data mining techniques such as PCA and NN. The results presented in the manuscript show an overall increase in water quality parameters (Chlorophyll, turbidity, and nitrogen concentration) with sur-

face area. Consequently, the conclusion confirmed these results by stating proportional relation between lake's water quality and surface area. In my opinion, the methodology and analysis presented to test the hypothesis are strong to reach to a solid conclusion. Therefore, I would recommend it for publication.

General comments: 1. The hypothesis of change in water quality with surface area is only applicable if there is no inflow and no outflow to/from the lake. Hence, the change in water level is only due to either evaporation or precipitation. Please explain. 2. Field data collection and validation are missing in the article. Please clarify. 3. Author relates surface area to each water quality index obtained by Sentinel-2 MSI. An average of all pixels (∼8 km2 of lake area) was considered as a potential match up of remotely estimated lake water surface area. Result shows an obvious increase with increase in water inflow (or lake' water level/surface area). Is it possible to divide the lake into different regions (based on water input/outputs) before applying analysis? 4. Basically, no green algae produce toxic gases, but cyanobacteria. Please check it.

Technical comments: Line 35 organic material not organic martial Line 37 organic material not organic martial Line 82 algae instead of Algae Line 82-83 "oxygen dissolving eutrophication process" Line 84 5 kilometers instead of 5 kilo meters

Yours sincelely,

Silvena Boteva

---

## Referee Comment (RC2) · Nese Yilmaz (Referee) · 20 Aug 2019

Remarks for Authors, The implementation of the partition analysis in the current article titled as "Effect of Water Surface Area on the Remotely Sensed Water Quality Parameters of Baysh Dam Lake, Saudi Arabia" is a fairly exercised and presented in a concise but thorough way and gives useful information. In my opinion, the article is worth publishing after the consideration of the following remarks:

Major remarks: why Sentinel 2, please explain. why 2 years only why 1 year is not enough or 4 years for example why 1 hidden layer and why 2 nodes, please explain.

Minor remarks: In page 1, line 12, its "of" instead of "on" In page 1, line 13, its water volume not water amount In page 1, lines 13-14, please rephrase In page 1, line 18, its and monitor instead of record In page 1, line 20, estimated from In page 1, lines 23-26, please rephrase to add more results In page 2, line 30, please start a new sentence In page 2, line, 32, please replace the word "big" In page 2, lines 32-34, kindly rephrase In gape 2, lines, 37-39, kindly rephrase In page 2, line 40-43, kindly omit In page 3, line 50, kindly rephrase In page 3, line 51, continuity of In page 3, lines 52-53, kindly rephrase In page 4, line 84, Kilometer is one word In page 4, line 92, kindly replace "sake" In page 4, lines 119-122, redundant text, kindly omit In page 9, line 184, what does it mean "in km" In page 11, line 191, kindly omit "s" confirm In page 11, line 193-196, kindly rephrase In page 12, line 207, is it 51 or 52

―――――――――――――――――――――

---

## Referee Comment (RC3) · Anonymous Referee #3 · 2 Sep 2019

This study develops potentially strong relationships between water surface area and satellite derived water quality indices in a reservoir. However, before the quality of this work can be properly assessed, the manuscript needs a thorough rewrite. There are numerous spelling mistakes and typos throughout the manuscript as well as the frequent use of ambiguous terminology within sentences making it difficult for a reader to follow the study. The authors need to provide additional details, particularly in the methodology and results sections, to demonstrate the importance of this study's findings as, currently, it appears to be an extension of a previous journal publication (Water 2019, https://doi.org/10.3390/w11030556). This is exemplified by Figure 1, which is directly referenced from this publication, and the reliance on field measurements col-

lected in the previous study.

Suggested areas for additional details are provided in the following sections.

Methodology considerations. Were any additional field measurements collected for this study, surface turbidity measurements across multiple sites within a single day would be of particularly interest to support any spatial patterns observed in satellite imagery? Please could the authors provide the location of field sampling sites in Figure 1. If any catchment inflows occurred, data such as particle size distribution, total suspended solids concentration or turbidity of inflow waters would be of great value to support any spatial patterns observed. Please could the authors provide a timeline of changes in reservoir surface area across the image collection period, noting times of any inflow events during this period. Were any water column light profiles collected from this reservoir during the image collection period, this would assist readers understand the variability in the underwater light field.

Results considerations. The central finding of increasing turbidity with decreasing surface area is a little difficult to reconcile with traditional reservoir behaviour where turbid inflows result in riverine sections of reservoirs experiencing higher turbidity during increases in surface area. Were any inflows captured by the satellite imagery, if so what spatial patterns in turbidity were observed? There is no spatial information provided in this work yet this seems to be an important advantage in using remote sensing of water quality, were any spatial water quality patterns observed over the image collection period? What are the potential drivers of surface turbidity in this reservoir, in systems that experience extended drawdown conditions wind driven resuspension of sediments in the shallow edges can be of importance in driving turbidity, could this be a possibility in this reservoir? Please could the authors provide the range of surface nitrogen, turbidity and chlorophyll concentrations observed in the field data, this would help readers that are not familiar with water quality in the study region.
* * *
308, 2019.

---

## Author Comment (AC1) · 2 Sep 2019

Reviewer #1 Comments and Suggestions for Authors The manuscript entitled "Effect of Water Surface Area on the Remotely Sensed Water Quality Parameters of Baysh Dam Lake, Saudi Arabia" by Elhag et al. presents a study based on the hypothesis that changes in surface area can affect water quality in Baysh Dam Lake. To evaluate this hypothesis, authors have used remotely estimated water surface area, water quality indices, and several statistical data mining techniques such as PCA and NN. The results presented in the manuscript show an overall increase in water quality parameters (Chlorophyll, turbidity, and nitrogen concentration) with face area. Consequently,

the conclusion confirmed these results by stating proportional relation between lake's water quality and surface area. In my opinion, the methodology and analysis presented to test the hypothesis are strong to reach to a solid conclusion. Therefore, I would recommend it for publication. General comments: 1. The hypothesis of change in water quality with surface area is only applicable if there is no inflow and no outflow to/from the lake. Hence, the change in water level is only due to either evaporation or precipitation. Please explain. The outflow of the lake operates only if the total volume of the dam lake water reaches more than at 80 BCM and is not common to have more than 80 BCM because the inflow is highly dependent on rainstorms not regular raining seasons. During the time frame of the current study (2 years, 2017-2018) there is no significant raining storms took place and mostly the water volume was slightly affected by evaporation process that we can neglect because it remained constant. the minimum surface area of the dam lake was 3.2 Km and the maximum was 4.8 km therefore the range of change (1.6 km) doesn't really support a potential inflow. Moreover, the findings of the current study remains valid as far as the minimum and the maximum water surface area were approximately endured the same. 2. Field data collection and validation are missing in the article. Please clarify. Data collection and data assessment were previously carried out and clearly cited in the current article. To avoid text redundancy data collection was omitted and the relevant reference was used as the follows: M Elhag, I Gitas, A Othman, J Bahrawi, P Gikas. 2019. Assessment of Water Quality Parameters Using Temporal Remote Sensing Spectral Reflectance in Arid Environments, Saudi Arabia. Water, 11(3):556-564. 3. Author relates surface area to each water quality index obtained by Sentinel-2 MSI. An average of all pixels (âĹij8 km2 of lake area) was considered as a potential match up of remotely estimated lake water surface area. Result shows an obvious increase with increase in water inflow (or lake' water level/surface area). Is it possible to divide the lake into different regions (based on water input/outputs) before applying analysis? Unfortunately it can't be done. Some parts of the lake are strictly prohibited because of the current war. So if we divided the lakes into regions there will be no validation to the restricted part. 4. Basically, no

green algae produce toxic gases, but cyanobacteria. Please check it. Acknowledged and corrected, its cyanobacteria Technical comments: Line 35 organic material not organic martial Acknowledged and corrected Line 37 organic material not organic martial Acknowledged and corrected Line 82 algae instead of Algae Acknowledged and corrected Line 82-83 "oxygen dissolving eutrophication process" Acknowledged and corrected Line 84 5 kilometers instead of 5 kilo meters Acknowledged and corrected

---

## Author Comment (AC2) · 2 Sep 2019

Reviewer #2 Remarks for Authors, The implementation of the partition analysis in the current article titled as "Effect of Water Surface Area on the Remotely Sensed Water Quality Parameters of Baysh Dam Lake, Saudi Arabia" is a fairly exercised and presented in a concise but thorough way and gives useful information. In my opinion, the article is worth publishing after the consideration of the following remarks: Major remarks: Why Sentinel 2, please explain. Data availability and the high sensitivity of the central wavelength (nm) of the Sentinel 2 bands excel the sensor to be used in related fields of qualitative and qualitative water studies. Why 2 years only why 1 year

is not enough or 4 years for example The minimum data input for the temporal analysis to be steady and gives reliable results shouldn't be less than 50 entries according to Psilovikos and Elhag 2013. The current study is taking into consideration the monthly data collection which 52 overall entries in the current study to make sure the results are robust and statically approved Why 1 hidden layer and why 2 nodes, please explain. 1 hidden layer is used to avoid the overfitting of the analysis which may lead to less trustful results. 2 nodes because we have only 3 parameters to consider and number of the used nodes are calculated as the number of the considered parameters -1. Minor remarks: In page 1, line 12, its "of" instead of "on" Acknowledged and corrected In page 1, line 13, its water volume not water amount Acknowledged and corrected In page 1, lines 13-14, please rephrase Acknowledged and rephrased In page 1, line 18, its and monitor instead of record Acknowledged and corrected In page 1, line 20, estimated from Acknowledged and corrected In page 1, lines 23-26, please rephrase to add more results Acknowledged and rephrased In page 2, line 30, please start a new sentence Acknowledged and started In page 2, line, 32, please replace the word "big" Acknowledged and replaced In page 2, lines 32-34, kindly rephrase Acknowledged and rephrased In page 2, lines, 37-39, kindly rephrase Acknowledged and rephrased In page 2, line 40-43, kindly omit Acknowledged and omitted In page 3, line 50, kindly rephrase Acknowledged and rephrased In page 3, line 51, continuity of Acknowledged and corrected In page 3, lines 52-53, kindly rephrase Acknowledged and rephrased In page 4, line 84, Kilometer is one word Acknowledged and replaced In page 4, line 92, kindly replace "sake" Acknowledged and replaced In page 4, lines 119-122, redundant text, kindly omit Acknowledged and omitted In page 9, line 184, what does it mean "in km" Acknowledged and explained In page 11, line 191, kindly omit "s" confirm Acknowledged and omitted In page 11, line 193-196, kindly rephrase Acknowledged and rephrased In page 12, line 207, is it 51 or 52 Acknowledged and corrected

---

## Author Comment (AC3) · 2 Sep 2019

This study develops potentially strong relationships between water surface area and satellite derived water quality indices in a reservoir. However, before the quality of this work can be properly assessed, the manuscript needs a thorough rewrite. There are numerous spelling mistakes and typos throughout the manuscript as well as the frequent use of ambiguous terminology within sentences making it difficult for a reader to follow the study. Acknowledged and improved The authors need to provide additional details, particularly in the methodology and results sections, to demonstrate the importance of this study's findings as, currently, it appears to be an extension of a previous

journal publication (Water 2019, https://doi.org/10.3390/w11030556). This is exemplified by Figure 1, which is directly referenced from this publication, and the reliance on field measurements collected in the previous study. Suggested areas for additional details are provided in the following sections. Methodology considerations. Were any additional field measurements collected for this study, surface turbidity measurements across multiple sites within a single day would be of particularly interest to support any spatial patterns observed in satellite imagery? Please could the authors provide the location of field sampling sites in Figure 1. It will be hard to show 120 sampling sites over the study area, we tried to layover the sampling sites over the study area but the result was a clumping figure of points. If any catchment inflows occurred, data such as particle size distribution, total suspended solids concentration or turbidity of inflow waters would be of great value to support any spatial patterns observed. There was no considerable catchment inflow Please could the authors provide a timeline of changes in reservoir surface area across the image collection period, noting times of any inflow events during this period. There was no considerable inflow Were any water column light profiles collected from this reservoir during the image collection period, this would assist readers understand the variability in the underwater light field. These suggestions are totally irrelevant to the study objectives. Results considerations. The central finding of increasing turbidity with decreasing surface area is a little difficult to reconcile with traditional reservoir behaviour where turbid inflows result in riverine sections of reservoirs experiencing higher turbidity during increases in surface area. Were any inflows captured by the satellite imagery, if so what spatial patterns in turbidity were observed? Again, there was not any considerable inflow to support your argument. Besides the dam supports the downward farms regardless the dam inflow which led to an extensive water withdrawn from the dam. Water shortage extends the degradation of the dam water quality. There is no spatial information provided in this work yet this seems to be an important advantage in using remote sensing of water quality, were any spatial water quality patterns observed over the image collection period? There wasn't a significant spatial varation to be mentioned, the lake is small to express a spa-

tial trend. Kindly bear in mind that most of the lake now is under siege because of the running war in south of Saudi Arabia. What are the potential drivers of surface turbidity in this reservoir, in systems that experience extended drawdown conditions wind driven resuspension of sediments in the shallow edges can be of importance in driving turbidity, could this be a possibility in this reservoir? Along the time frame of the current study, the lake of the dam didn't receive and remarkable inflows which significantly affect the dam water quality. Moreover, the dam supports the downward farms regardless the dam inflow which led to an extensive water withdrawn from the dam. Water shortage extends the degradation of the dam water quality. Please could the authors provide the range of surface nitrogen, turbidity and chlorophyll concentrations observed in the field data, this would help readers that are not familiar with water quality in the study region. Acknowledged and added

————————————————————